

# Satellite-derived sulphur dioxide (SO$_2$) emissions from the 2014-2015 Holuhraun eruption (Iceland)

Elisa Carboni[1], Tamsin A. Mather[2], Anja Schmidt[3,4], Roy G. Grainger[1], Melissa A. Pfeffer[5], and Iolanda Ialongo[6]

[1]COMET, Atmospheric, Oceanic and Planetary Physics, University of Oxford, Clarendon Laboratory, Parks Road, Oxford OX1 3PU, UK.
[2]COMET, Department of Earth Science, University of Oxford, South Park Road, Oxford OX1 3AN, UK.
[3]Department of Chemistry, University of Cambridge, Cambridge, CB2 1EW, UK
[4]Department of Geography, University of Cambridge, Downing Place, Cambridge CB2 3EN, UK
[5]Icelandic Meteorological Office, Bustadavegur 7-9, Reykjavik, Iceland
[6]Space and Earth Observation Centre, Finnish Meteorological Institute, Helsinki, Finland

**Correspondence:** Elisa Carboni (elisa.carboni@physics.ox.ac.uk)

**Abstract.** The six-month-long 2014-2015 Holuhraun eruption was the largest in Iceland for 200 years, emitting huge quantities of sulphur dioxide (SO$_2$) into the troposphere, at times overwhelming European anthropogenic emissions. Weather, terrain and latitude, made continuous ground-based or UV satellite sensor measurements challenging. Infrared Atmospheric Sounding Interferometer (IASI) data, is used to derive the first time-series of daily SO$_2$ mass and vertical distribution over the eruption

period. A new optimal estimation scheme is used to calculate daily SO$_2$ fluxes and average e-folding time every twelve hours. The algorithm is used to estimate SO$_2$ fluxes of up to 200 kt per day and a minimum total SO$_2$ erupted mass of $4.4\pm0.8$ Tg. The average SO$_2$ e-folding time was $2.4\pm0.6$ days. Where comparisons are possible, these results broadly agree with ground-based near-source measurements, independent remote-sensing data and model simulations of the eruption. The results highlight the importance of high-resolution time-series data to accurately estimate volcanic SO$_2$ emissions. The dataset derived can be used

for comparisons to other ground- and satellite-based measurements, and to petrological estimates of the SO$_2$ flux as well as to initialize climate models, helping to better quantify the environmental and climatic impacts of flood lava eruptions.

## 1   Introduction

Sulphur dioxide (SO$_2$) is one of the most important magmatic volatiles for volcanic geochemical analysis and hazard assessments due to its low ambient concentrations, abundance in volcanic plumes and spectroscopic features. Tropospheric volcanic

SO$_2$ and its conversion products can affect the environment, human health, air quality and the radiative balance of the Earth (Gíslason et al., 2015; Schmidt et al., 2012; Gettelman et al., 2015; Ilyinskaya et al., 2017; Boichu et al., 2016). Measurements of SO$_2$ from volcanic eruptions are vital both to understand the underlying volcanic processes and also the wider scale environmental impacts of volcanism. The Icelandic Holuhraun eruption lasted from 31 August 2014 to 28 February 2015 and produced the largest lava volume in Iceland for more than 200 years (Gíslason et al., 2015). During September 2014, Holuhraun's average

daily SO$_2$ emission exceeded daily SO$_2$ emissions from all anthropogenic sources in Europe by a factor of three (Schmidt et al.,





2015 and references therein). The weather conditions and terrain made regular ground-based plume measurements extremely challenging during the winter months (Pfeffer et al., 2018). The high latitude of the eruption meant there was insufficient sunlight to reliably detect the volcanic plume using UV satellite sensors beyond the end of the October 2014 and ground based UV instruments did not measure $SO_2$ during the darkest seven weeks of winter. Under these circumstances satellite-based thermal

infrared spectrometers are an optimal source of high temporal resolution $SO_2$ amount and altitude.

The Infrared Atmospheric Sounding Interferometers (IASI) on-board the Metop satellite platforms provide several observations of Holuhraun each day. The plume altitude and $SO_2$ column amount are retrieved from the measured top-of-atmosphere spectral radiance (Carboni et al., 2012). For the first month of the Holuhraun eruption, previous studies have shown good agreement between IASI measurements and those from the Ozone Monitoring Instrument (OMI), ground-based and balloon-borne

measurements, and atmospheric dispersion model simulations (Schmidt et al., 2015; Vignelles et al., 2016). In this work IASI measurements are used to produce the first time series of the Holuhraun $SO_2$ plume. Retrievals of $SO_2$ amount from Metop-A satellite are binned and averaged for successive 12 hour periods to give coverage for the entire period of the eruption. The time-series of the $SO_2$ mass present in the atmosphere is used to calculate $SO_2$ fluxes and an average $SO_2$ e-folding time, under the assumption that the flux is constant over a twelve hour period. The results are compared with ground-based Brewer

measurements of the $SO_2$ column amount and with measurements of near-source plume altitudes and fluxes from the Icelandic Meteorological-Office (IMO). The dataset presented can be used for comparisons to other ground- and satellite-based measurements and to petrological estimates of the $SO_2$ flux and to initialise, for instance, climate model simulations, helping to better quantify the environmental and climatic impacts of volcanic $SO_2$.

## 2   IASI $SO_2$ iterative retrieval scheme

IASI is an infrared Fourier transformer interferometer, on-board the Metop-A and Metop-B satellites. It measures in the spectral range 645-2760 $cm^{-1}$ with spectral sampling of 0.25 $cm^{-1}$ and has global coverage every 12 hours. The IASI data used in this study was the level 1b dataset from the EUMETSAT and CEDA archive.

The details of the retrieval scheme are summarized briefly below. For more details see Carboni et al. (2012, 2016). An $SO_2$ retrieval is performed for all IASI pixels that present a positive result in the $SO_2$ detection scheme (Walker et al., 2011,

2012). The detection scheme uses all the channels in the range 1300-1410 cm-1 ($\nu_3$ band). The detection limits for a standard atmosphere (with no thermal contrast) are estimate to be: 17 DU for a $SO_2$ plume between 0-2 km, 3 DU between 2-4 km and 1.3 DU between 4-6 km (Walker et al., 2011). The detection scheme can miss part of an $SO_2$ plume under certain circumstances, such as low-altitude plumes, conditions of negative thermal contrast (i.e. where the surface is colder than the atmosphere), and where clouds are present above the $SO_2$ plume, masking the signal from the underlying atmosphere. Due to

these uncertainties the estimated mass of $SO_2$ in this paper should be regarded as a 'minimum'.

All the channels in the ranges 1000-1200 and 1300-1410 $cm^{-1}$ (the 7.3 and 8.7 $\mu$m $SO_2$ bands) are simultaneously used in the iterative optimal estimation retrieval scheme to obtain the $SO_2$ amount, the altitude of the plume and the surface tempera-



ture. The $SO_2$ band around 8.7 $\mu m$ (1000 to 1200 cm$^{-1}$) is within an atmospheric window. This allows the radiation from the surface to reach the satellite from deep within the atmosphere enabling the retrieval of $SO_2$ amount down to the surface.

A comprehensive error budget for every pixel is included in the retrieval. This is derived from an error covariance matrix that is based on the $SO_2$-free climatology of the differences between the IASI and forward modeled spectra. Rigorous error
propagation, including the incorporation of forward model and forward model parameter error, is built into the system, providing quality control and comprehensive error estimates on the retrieval results. The IASI $SO_2$ retrieval is not affected by underlying cloud. If the $SO_2$ is within or below a cloud layer its signal will be masked and the retrieval will underestimate the $SO_2$ amount.

## 3  Temporal evolution of $SO_2$ mass and $SO_2$ vertical distribution

The retrieval results from the Metop-A orbits during the period from September 2014 to February 2015 and from 30° N to 90° N are combined (twice a day, i.e. morning and afternoon overpasses) to produce maps of retrieved $SO_2$ amount and altitude. Supplementary movie S1 shows the evolution of the plume for each day. The Holuhraun eruption is the main source of $SO_2$ over that period. Other minor sources include $SO_2$ emitted from intermittent volcanic activity on the Kamchatka peninsula, the Etna volcano (28[th] December 2014, 1[st], 2[nd] and 21[st] January 2015), and anthropogenic $SO_2$ emissions from China/Beijing. Satellite
observations at the pixel level do not provide sufficient information to distinguish between $SO_2$ from Holuhraun and $SO_2$ from other sources. For example, the elevated $SO_2$ near Beijing on 21[st] December 2014 appears to be from an anthropogenic source, but the elevated $SO_2$ in the same area on 31[st] December 2014 is from the Holuhraun eruption. In this study all the $SO_2$ measured from 30° N to 90° N between September 2014 and February 2015 is referred to as Holuhraun $SO_2$.

Over the course of the six months the eruption plume dispersed across the Northern Hemisphere. Figure 1 shows the max-
imum $SO_2$ column amount retrieved during the six-month period and illustrates that $SO_2$ from the Holuhraun eruption was dispersed over large parts of the Northern Hemisphere including poleward of the Arctic circle. For the majority of the time the plume circulated around the pole and the northern regions (see animation in Supplementary Material Figure SX), overpassing Scandinavia, Eastern Europe, Russia, Greenland and Canada several times. The plume overpassed Europe on multiple occasions, most often northern Europe (Schmidt et al., 2015; Ialongo et al., 2015; Zerefos et al., 2017; Steensen et al., 2016;
Twigg et al., 2016), but also Italy (22nd October 2014) and Spain and reaching as far south as Morocco and Algeria (on 5[th] and 6[th] November 2014 respectively) and Greece/Macedonia/Albania (5[th] and 6[th] January 2015).

The $SO_2$ mass present in the atmosphere for each IASI overpass was found by regridding the observations of column amount and plume altitude into a 0.125° latitude/longitude boxes following Carboni et al. (2016). The $SO_2$ mass time-series is obtained by summing the mass values of the regularly gridded map for each 12 hour period. The time-series of $SO_2$ mass,
together with the errors, are presented in the top plot of Figure 2. The $SO_2$ mass is highest in September 2014 (up to 0.25 Tg) when the eruption was most powerful. The $SO_2$ mass decreases during October 2014 (with some peak values around 0.1 Tg) then increases around end of November/beginning December 2014 (up to 0.15 Tg). The $SO_2$ mass steadily declines during



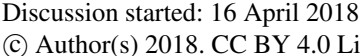

**Figure 1.** Map showing the maximum of $SO_2$ column amount (in Dobson Units, DU) retrieved within the considered area from $30°$ N to $90°$ N (black rectangle) from September 2014 to February 2015.

January and February 2015 as the eruption comes to an end. There is no detection of $SO_2$ plume attached to the vent in the second half of February (and the $SO_2$ mass for this period, reaching a value up to $0.01$ Tg, is from a non-Icelandic source).

The $SO_2$ mass present between two altitude levels was estimated using the method of Carboni et al. (2016) to produce the vertical distribution of $SO_2$. In this study the vertical distribution of $SO_2$ was estimated every 12 hours from 0 and 10 km with a vertical resolution of 0.5 km for all latitudes north of $30°$ N. Both the young emitted plume as well as the mature plume that had been transported around in the Northern Hemisphere for few days are included in the distribution. The time-series of the $SO_2$ vertical distribution for the Holuhraun eruption is shown in Figure 3. The centre-of-mass of the plume closest to the vent can be used as rough estimate for the injection height. Figure 3 shows the time-series of two datasets: (i) the vertical distribution and (ii) the altitude of the centre-of-mass of the $SO_2$ values within 500 km of the vent. The altitude of the centre-of-mass is less than 4 km for the majority (96%) of the measurements.



**Figure 2.** Time-series of the masses and emission fluxes of SO₂ as function of day from 1st January 2014. Top plot shows the IASI masses with error-bars in black and the fitting of the retrieval with the blue line. The bottom plot shows the retrieved SO₂ flux time-series from IASI in black with grey error-bars and from IMO using DOAS in red (red bars show the errors, dotted bars show the maximum and minimum values measured that day).



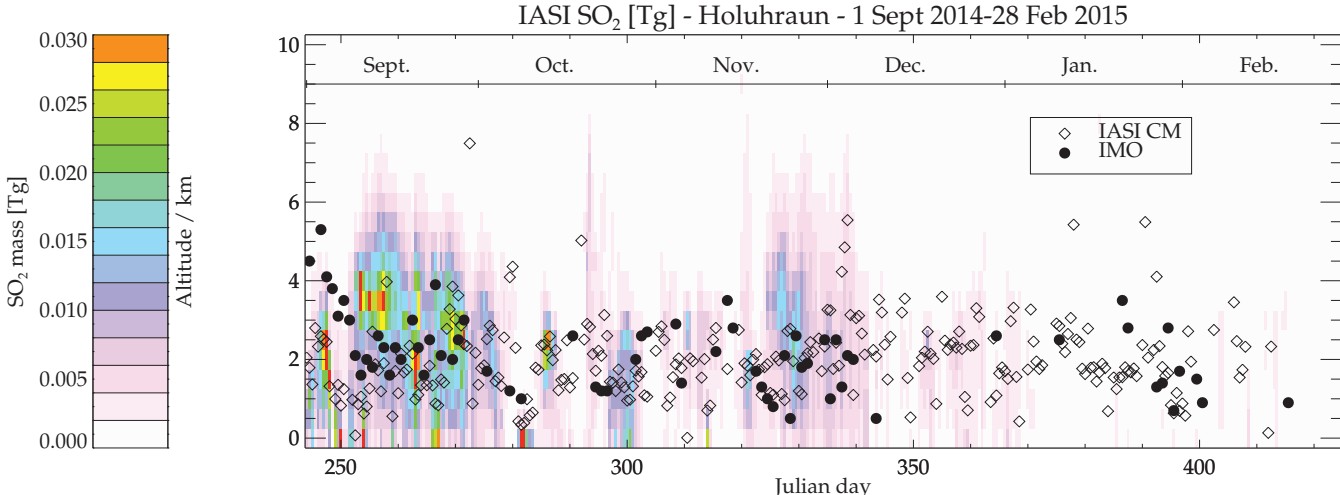

**Figure 3.** $SO_2$ vertical distribution in km above sea level. The colour represents the mass of $SO_2$, dark-red represents values higher than the colour-bar. Every column of the plot is generated from an IASI map (one every 12 hrs; Figure 1 and supplementary files). The black diamonds show the altitude of the centre of mass computed with the IASI pixels within 500 km from the vent, the black dots show the altitude from IMO measurements.

## 4 Daily $SO_2$ fluxes

The time-series of $SO_2$ fluxes and the coefficient of an average exponential decay of $SO_2$ (the state vector $\mathbf{x}$) are retrieved
5 from the time-series of $SO_2$ mass (the measurement vector $\mathbf{y}$) using the optimal estimation scheme of Rodgers (2000). The solution ($\mathbf{x}$, the vector of parameters that we want to retrieve) gives the most probable $\mathbf{x}$ given the measurements y and the a priori knowledge $\mathbf{x}_a$. The state vector $\mathbf{x}$ is found by minimizing the cost function:

$$\chi^2 = [\mathbf{y} - \mathbf{F}(\mathbf{x})]^T \mathbf{S}_e^{-1} [\mathbf{y} - \mathbf{F}(\mathbf{x})] + [\mathbf{x} - \mathbf{x}_a]^T \mathbf{S}_a^{-1} [\mathbf{x} - \mathbf{x}_a] \tag{1}$$

where $\mathbf{F}$ is the forward model (that simulates the measurement given the state vector), $\mathbf{x}_a$ is the a priori value of the state vector and $\mathbf{S}_e$ and $\mathbf{S}_a$ are the measurement and a priori error covariance matrices. $\mathbf{S}_e$ and $\mathbf{S}_a$ are diagonal matrixes with the variances
10 (square of errors) of $\mathbf{y}$ and $\mathbf{x_a}$ respectively as their diagonal elements. The a priori values used were $0.2 \pm 0.2$ Tg/day for flux and $2 \pm 2$ day for the e-folding time. To retrieve the time-series of $SO_2$ fluxes the state vector $\mathbf{x}$ was defined as follows. The first element of the state vector is the average $SO_2$ e-folding time ($\lambda$) for the period analysed; the following $x_i$ elements are the average emission flux $f_i$ between the two IASI estimates of $SO_2$ mass at time $t_{i-1}$ and $t_i$:

$$\mathbf{x} = (\lambda, f_1, f_2, ..., f_n) \tag{2}$$

To define the forward model $\mathbf{F}$ we consider that the $SO_2$ mass $m$ measured by satellite is a function of time and that the mass decays proportionally to the mass itself, plus the addition of a source term of flux $f$. These terms give a first order differential





equation:

$$\frac{dm}{dt} = -km + f \tag{3}$$

Assuming a constant flux $f$ over the time interval $\Delta t$ between two consecutive mass estimates $m_i$ and $m_{i-1}$, the solution

becomes:

$$m_i = m_{i-1}e^{-\frac{1}{\lambda}\Delta t} + f\lambda(1 - e^{-\frac{1}{\lambda}\Delta t}) \tag{4}$$

where $\lambda$ (with $\lambda = 1/k$) is the average e-folding time. Equation 4 is the forward model $\mathbf{F}(\mathbf{x})$.

Figure 2 shows the retrieved fluxes with errors. The flux time-series follows a similar pattern to the $SO_2$ mass time-series, with higher values in September 2014. IASI determined fluxes are provided here with the ground-based measurement fluxes provided in parenthesis (described in section 5). The September daily average was 0.06 Tg/day (0.09) and the September daily

maximum was 0.26 Tg/day on the 20[th] September (0.19). The October average flux was 0.023 Tg/day (0.083), with peak values of 0.15 and 0.12 Tg/day on the 11[th] and 19[th] of October (0.10). The November average flux was 0.029 Tg/day (0.069). The flux increases in the second part of November to a maximum of 0.13 Tg/day on the 23[rd] of November (0.13). The estimates for December, January and February show decreasing flux with monthly average of 0.016, 0.006 and 0.005 Tg/day respectively (0.026, 0.028, 0.016). The monthly averages are lower than those measured by the ground-based measurements while the

maximum daily averages for each month are generally higher. The fluxes calculated for September 2014 are consistent with Schmidt et al. (2015) (e.g. up to 0.120 Tg/d during early September, 0.02-0.6 Tg/day between the 6[th] and 22[nd] of September, 0.06-0.120 Tg/day until the end of September).

The total mass of $SO_2$ emitted by the eruption is obtained by summing all the fluxes $f_i$, output by the retrieval and multiplying by the corresponding $\Delta t_i$. The error associated with the total mass emitted is obtained by adding in quadrature the errors $\delta f_i$

multiplied by the time interval $\Delta t_i$. The maximum value of total mass emitted is obtained by summing all the fluxes plus the uncertainty and the minimum value is obtained by summing all the fluxes less the uncertainty (negative values are set to zero). This gives a total mass of emitted $SO_2$ of $4.4 \pm 0.8$ Tg with a maximum of 11.6 Tg and a minimum of 0.4 Tg. The retrieved averaged e-folding time is $2.4 \pm 0.6$ days.

Note that in using an average $SO_2$ e-folding time for the entire eruption period, any variation of e-folding time will be

interpreted as an inverse variation in the estimated flux, i.e when the 'real' e-folding time is higher than the retrieved one, the flux will be overestimated and vice versa. The flux uncertainties include the errors in flux due to the variation in e-folding time. The $SO_2$ lifetime can vary significantly on a daily basis mainly as a function of water vapour and solar irradiation. Also note that any e-folding time shorter then the retrieved one can fit the measurements and give higher fluxes. Given these caveats the value of e-folding time ($2.4 \pm 0.6$ days) is consistent with the mean lifetime of 2.0 days estimated for September

2014 (Schmidt et al., 2015). Their estimate was based on minimising the difference between the $SO_2$ amount from the NAME dispersion model and the IASI and OMI satellite measurements.

Figure 2 shows that higher values (peaks) of $SO_2$ flux often alternate with lower values (below 0.02 Tg/day) even during periods that have been identified as generally characterized by higher fluxes. This intermittent flux behaviour has important





implications in terms of the estimate of total $SO_2$ emitted. Had a less sensitive instrument been used that only produced 'valid' measurements in correspondence with higher flux values (e.g. > 0.05 Tg/day), and had considered these fluxes as representative

of the period without valid measurements (i.e. period between two 'valid' measurements), this would result in very different (and higher/overestimated) estimated values of total $SO_2$ emitted. An example of this is Gauthier et al. (2016) where they used TIR data from SEVIRI, on board the geostationary satellite Meteosat Second Generation (MSG), to retrieve an $SO_2$ mass time-series from 1 September 2014 to 25 November 2014. Their retrieved mass values are lower compared to the IASI values here (due to the smaller geographic area considered and possibly due to a smaller sensitivity or detection threshold of

SEVIRI), nevertheless they estimate a total $SO_2$ emitted mass of $8.9 \pm 0.3$ Tg for the period September 2014 to November 2014, which is a factor of two higher than calculated here. The ground-based measurements reported in Pfeffer et al. (2018) show an intermediate value of 7.3 Tg over this time interval.

## 5   Comparison with ground-based and near-source measurements

The conditions of the Holuhraun eruption are significantly different to eruptions investigated in previous studies using IASI

(Carboni et al, 2016), because the plume from Holuhraun was confined to altitudes between the surface and 6 km at high-northern latitudes and because the eruption took place during the autumn and winter months. As a result there is less radiance and low (or negative) thermal contrast between the surface and the first layer of the atmosphere. These conditions lead to lower sensitivity for measurements in both the UV and TIR spectral range. Nevertheless a cross-comparison with other available measurements is informative when assessing our results. The following comparisons are an addition to previous comparison

done with UV satellite and dispersion model (Schmidt et al., 2015) and balloon measurement (Vignelles et al., 2016).

First the IASI dataset is compared with ground-based Brewer measurements of the $SO_2$ column amount of the mature plume over Finland and then the plume height is compared with near-source measurements in Iceland using ground- and aircraft-based visual observations, web camera and NicAIR II infrared images, triangulation of scanning DOAS instruments, and the location of $SO_2$ peaks measured by DOAS traverses as reported in Pfeffer et al. (2018).

The Brewer ground measurements (Ialongo et al., 2015) were made at Sodankylä (67.42° N, 26.59° E). The $SO_2$ column amounts are routinely obtained from the direct solar irradiances at wavelengths of 306.3, 316.8 and 320.1 nm, by using the same Brewer algorithm as for the ozone retrieval (Kerr et al., 1988). The method is based on the Lambert-Beer law, which describes the attenuation of the direct solar irradiance reaching the Earth's surface at certain wavelengths due to the atmospheric constituents. In order to avoid the effects of stray light at short wavelengths, the measurements corresponding to large air

mass values (after 14:20 UT) are not included. The $SO_2$ column amounts in Sodankylä are typically close to zero with an estimated detection limit of about 1 DU. Ialongo et al. (2015) compared the $SO_2$ column amount values from Brewer and OMI measurements in Sodankylä during September 2014 with differences between OMI and Brewer retrievals were usually smaller than 2 DU.

The comparison here is performed by averaging all the IASI pixels that pass quality control, within 200 km of the ground measurements. The Brewer instrument measures in the UV and thus only in daylight conditions. This means that only the first



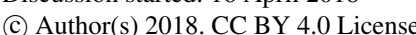

month of the eruption can be considered. Fig.4 shows the time-series of $SO_2$ column amount from both ground measurements
and IASI as function of time. All the 'plume' episodes (with $SO_2$ amount larger than 2 DU) are consistent between the two

5    datasets with the exception of 15th and 19th September where the plume only passes over the northern part of the 200 km
circle in the IASI data and does not pass over the ground measurement station. This means that while IASI presents high $SO_2$
measurements, elevated values are not observed in the Brewer measurements.

**Figure 4.** Time-series of the $SO_2$ column amount [DU] measurements at Sodankylä as function of day from the 1st September 2014. Black
symbols are the Brewer measurements with error bars, red symbols are the mean and standard deviation of all the IASI measurements within
200 km, blue lines represent the maximum and minimum of the IASI measurements.

There are a few days of low (less the 2 DU) $SO_2$ reported by only one of the two instruments (IASI or Brewer), meaning that
the detection limit of one of the instruments is not exceeded. This is consistent with the IASI minimum error for low altitude
plume (2 DU for plumes below 2 km, Carboni et al., 2016) and with the Brewer minimum error (2 DU, Sellitto et al., 2017).





The ground-based measurements of eruption cloud top height were collated from multiple techniques including ground- and aircraft-based observations, web camera, ScanDOAS, MobileDOAS, and NICAIR II IR camera (Pfeffer et al., 2018). The ScanDOAS and MobileDOAS approaches, and IASI retrievals, provide the height of the center-of-mass of the plume while the other techniques provide the height of the top of the plume. Fig 3 presents the ground-based and IASI altitudes together. In general, the IASI and ground-based altitudes agree that the altitudes varied mainly between 1-3 km, however they do not agree particularly well on any specific day.

The time-series of the ground-based (Pfeffer et al., 2018) and IASI flux measurements are presented in Figure 2. Within error, they generally agree. There are a few significant differences between the two datasets: two values in October/November 2014 and some values at the end of January and February. The ground-based measurements in November alternate between very high and very low values. On 5[th] November 2014, the ground-based value is significantly higher than the IASI flux estimate for that day. Pfeffer et al. (2018) suggest the high values in November could be due to degassing from a continually replenishing lava lake contributing to the total gas in addition to the degassing from the magma being erupted. The plume altitude was less than 2 km on this day and under these conditions IASI values can be underestimates. The total mass emitted can be estimated using the ground-based measurements as the integral below the red line in Figure 2 (i.e. interpolating flux values). Even if the majority of fluxes are consistent with each other within the error estimate, the total mass calculated by IASI (4.4 Tg) and IMO (9.6 Tg) differ by a factor of two. The 5[th] November discrepancy contributes significantly to this difference.

## 6 Conclusions

The first satellite-based $SO_2$ flux dataset of the full 2014-2015 Holuhraun eruption has been estimated using the IASI instruments. The dataset provides a flux estimate every 12 hours for the entire eruption. Thermal infrared spectrometers are the only satellite instruments that could follow the $SO_2$ plume around the Arctic in the absence of solar irradiation during the winter months of the eruption. The low-altitude of the $SO_2$ plume and cold underlying surface reduce IASI's sensitivity to $SO_2$, however the results compare resonably with ground-based near and distal measurements. The observations show that the Holuhraun plume passed over large parts of the northern hemisphere during the eruption. The time-series of $SO_2$ vertical distribution showed a low-altitude plume confined mainly within 0-6 km. The time-series of $SO_2$ masses showed a maximum of 0.25 Tg of atmospheric loading in September 2014. A new optimal estimation scheme was developed to calculate daily $SO_2$ fluxes and e-folding time based on satellite-retrieved atmospheric $SO_2$ burdens. Application of the method gave estimates of $SO_2$ flux of up to 200 kt/day. The 'minimum' total mass of $SO_2$ was calculated to be 4.4±0.8 Tg and the average $SO_2$ e-folding time was found to be $2.4 \pm 0.6$ days.

*Data availability.* The $SO_2$ data presented in this paper is available from the corresponding author on request.



*Competing interests.* The authors declare that they have no conflict of interest.

*Acknowledgements.* E. Carboni, R. G. Grainger and T. A. Mather were supported by the NERC Centre for Observation and Modelling of Earthquakes, Volcanoes, and Tectonics (COMET).



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
