# Peer review of "Satellite-derived sulphur dioxide (SO2) emissions from the 2014-2015 Holuhraun eruption (Iceland)"

_Atmospheric Chemistry and Physics, 2018_

## Referee Comment (RC1) · Anonymous Referee #1 · 17 May 2018

In the manuscript entitled "Satellite-derived sulphur dioxide (SO2) emissions from the 2014-2015 Holuhraun eruption (Iceland)", the authors derive the first timeseries of the SO2 emissions for the entire Holuhraun eruption. In a first stage, using a retrieval scheme previously developed, the authors retrieve the SO2 amount and altitude of the Holuhraun plume from IASI observations. Based on these, the authors then determine SO2 total masses every 12 hours in a large box (30°N-90°N) covering the Northern Hemisphere. They finally retrieve the SO2 fluxes for 12-hour periods using an optimal estimation scheme, considering the retrieved total masses as the measurement vector. To assess their results, the authors compare the retrieved SO2 columns, plume altitude and SO2 emissions with different type of ground-based measurements. While the SO2

emissions determined in this paper are important for different applications, a few main issues and several specific comments should be addressed before publication.

Major comments

1) Page 2, lines 27-30, the authors mention that because of some geophysical conditions, part of the SO2 plume can be missed by IASI, and thus, the derived SO2 masses should be considered as minima. This is totally true, the presence of clouds and/or low thermal contrast can hamper the detection of the SO2, and this is a complicated problem to deal with when estimating the SO2 masses. However, the authors stay very qualitative on this problem and more particularly do not mention this problem anymore in the rest of the paper (e.g. in the comparisons). It seems that the SO2 total mass derived by the authors for the entire eruption is lower than those previously estimated (Gauthier et al, 2016; Pfeffer et al., 2018; Gíslason et al., 2015; Thordarson and Hartley, 2015), but this is not discussed in terms of the underestimation of the SO2 masses and its effect on the estimated fluxes. Since the latter will be available for future comparisons and for model simulations, the authors should discuss this deeper and try to evaluate (and I realize that it is a complicated problem) how large can be the underestimation of the retrieved SO2 masses and how this underestimation affects their SO2 fluxes.

2) Page 3, lines 10-18, it is explained that the estimation of the SO2 masses is performed for a box going from 30°N to 90°N, considering that the SO2 detected comes from the Holuhraun eruption only. However, in this large box, other SO2 sources (China, Norilsk, volcanoes) are located and contribute to the total SO2 in this box. While the SO2 amount emitted by these sources is probably negligible compared to the one emitted at the beginning of the eruption (masses of 0.1-0.3 Tg), I am afraid these sources could contribute to a larger percentage the days where SO2 masses lower than 0.1 Tg are estimated. For instance, the annual SO2 emissions of Norilsk are estimated to be around 2 Tg (Fioletov et al., 2016). On a daily basis, this can lead to SO2 masses around 0.001-0.01 Tg, and this can represent a large percentage of

the estimated SO2 masses. This is especially the case from December, when the SO2 masses are mostly lower than 0.1 Tg. Moreover, in this period, the thermal contrast values and humidity conditions in the Norilsk area, but also in China, were shown to favour the measurement of near-surface SO2 (Boynard et al., 2014; Bauduin et al., 2014; 2016). How did the authors take into account these extra sources? Did they remove a background of SO2 from their masses? How large do they estimate the contribution of the other sources and how does it affect the estimated SO2 masses and fluxes? This issue deserves more explanations and investigations.

3) I have a few comments on the method used to estimate the SO2 fluxes. First of all, the authors should explain the advantages of the method they use compared to others ones (Theys et al., 2013). Then, I am concerned about the assumption of an averaged constant lifetime for the entire eruption. As mentioned by the authors, the lifetime of SO2 is very variable and depends on humidity, solar irradiation and altitude of the plume. Because the eruption lasted 6 months and the plume travelled very far from the source, these conditions significantly varied during the eruption and according to the location of the plume. Therefore, I am wondering why the authors have made this choice of method and why they did not consider a more sophisticated method, using a dispersion model to estimate the SO2 fluxes (Theys et al., 2013). At page 7, line 26, the authors mention that the flux uncertainties include the possible variation of the e-folding time. This is not clear how this is done. In conclusion, the authors should justify their choice of method and provide a clear explanation of how they assess the impact of a constant lifetime on the retrieved SO2 fluxes.

4) Regarding among other things the previous comments, the present paper lack references, especially related to the estimation of volcanic SO2 emissions and the SO2 lifetime. The following references should be at least added: McCoy and Hartmann, GRL 42, 10409-10414, 2015, doi:10.1002/2015GL067070; Malavelle et al., Nature 546, 485-491, 2017, doi:10.1038/nature22974; Lee et al., JGR 116, 2011, doi :10.1029/2010JD014758; Theys et al., ACP 13, 5945-5968, 2013, doi

[Figure]

:10.5194/acp-13-5945-2013; Carn et al., J. Volc. Geoth. Res. 311, 99-134, 2016, doi:10.1016/j.jvolgeores.2016.01.002.

5) Some parts of the manuscript are difficult to read and are not clear. For instance, the following paragraphs could be improved (see also specific and technical comments): Page 3, lines 3-6; Page 7, lines 24-31; Page 8, lines 1-12.

Specific comments

-Abstract, line 5: the use of the optimal estimation to infer the SO2 emissions is not something new (Theys et al., 2013).

-Abstract, line 6: "The algorithm is used to estimate SO2 fluxes of up to 200 kt...". This sentence sounds weird to me. I understand that the algorithm cannot estimate fluxes larger than 200 kt.

-Abstract, line 8: you say that you compared your results to model simulations. Do you refer to the comparison with the work of Schmidt et al. (2015)? You should rephrase the sentence because, the way it is written, the reader understands that you do model simulations.

-Page 2, line 12: can you specify what coverage? Is it temporal, spatial or both?

-Page 2, line 22: did you use IASI data of level 1b (not apodized)? Or 1c (obtained after apodization)?

-Page 2, line 24: Can you briefly remind what is a positive result in the SO2 detection scheme?

-Page 2, line 28: I would specify that you miss part of low-altitude SO2 plumes in case of low thermal contrast. On the same line, you say that IASI can miss part of the SO2 plume in case of negative thermal contrast conditions. Why? It has been shown by Bauduin et al. (2014, 2016) and Boynard et al. (2014) that negative thermal contrasts are favourable conditions to measure SO2 close to the surface.

-Page 3, line 2: Why cannot you use the $\nu 3$ band to measure SO2 down to the surface? Is the band saturated? Or is it because of large humidity close to the surface? Can you please explain? If none of these two reasons is true, you should be able to measure SO2 close to the surface using the $\nu 3$ band (Bauduin et al., 2014; 2016).

-Page 3, line 3: How did you build the error covariance matrix? Is it a global one or did you build one more specific for the eruption? This should be explained.

-Page 3, line 6: Can you specify what is your quality control?

-Page 3, line 7: You say that the SO2 retrieval is not affected by an underlying cloud. However, this cloud has to be taken in the retrieval, at least in the radiative transfer. How did you take into account the underlying clouds? How did you detect underlying clouds?

-Page 3, line 11: it is not clear to me what you combined. I suppose you created AM and PM maps each day?

-Page 3, lines 15-18: you say that you cannot make the distinction between the Holuhraun plume and the other SO2 sources, but then you make the distinction for the 21st and the 31st December. How did you do this distinction? How did you take this into account in the evaluation of the SO2 masses and fluxes? (major comment 2)

-Page 3, line 24: the reference Boichu et al. (2015) should be added.

-Page 3, line 30: can you specify how you calculate the errors on the total masses?

-Page 6, lines 10-11: why did you choose these a priori values? Did you rely on the previous literature?

-Page 7, line 7: you did not explain what Se you considered. Can you specify it?

-Page 7, line 13: The averaged fluxes reported for December, January and February are very low. The impact of other sources is in this case non negligible (major comment 2). Did you take this into account?

-Page 7, lines 14-15: You did not give a tentative explanation for the fact that 1) the monthly averages of IASI fluxes are lower than those calculated from ground-based observations, and 2) the maximum values are larger for IASI than for ground-based observations. Is this because of the underestimation of the masses? Or variations in the lifetime? Or the inclusion of other sources? I think you can extend the discussion.

-Page 7, line 16: You compare the SO2 fluxes you determined with the modelled fluxes of Schmidt et al. (2015). Why didn't you also compare your emissions with the fluxes they determined from OMI and IASI observations?

-Page 7: I would add the errors of the fluxes in the text.

-Page 7, line 18: you calculate the total mass of the eruptions from the SO2 emissions you derived. These emissions are strongly affected by the fact that you use a constant averaged SO2 lifetime. Wasn't it more accurate to use the daily masses you estimated (even though they are underestimated)?

-Page 7, lines 32-33: the "spikes" you see in the SO2 fluxes, are they real? Or do they come from the forward model you used? The Delta-M method is known to produce spikes in time-dependent fluxes (Theys et al., 2013).

-Page 8, lines 8-10: How do you explain that Gauthier et al. (2016) estimated lower SO2 daily masses but a higher total SO2 mass? Is this related to their choice of lifetime? This comparison should be discussed deeper.

-Page 8, lines 11-12: Following the previous comment, I think the comparison could be extended. You did not compare the total mass you estimated with those reported by Schmidt et al. (2015) (2 Tg for September 2014) and by Gíslason et al. (2015) (11 Tg). Moreover, you should compare the total masses calculated for a same period. The comparison should mention the difference in sensitivity, in lifetime,...

-Page 8, lines 17-18: As already mentioned above, negative thermal contrasts have been shown to increase the sensitivity to near-surface SO2 (Bauduin et al, 2014; 2016;

Boynard et al., 2014).

-Page 8, line 34: Can you explain why you compare ground-based measurements with the average of all IASI pixels located within 200 km? Why not taking the closest IASI pixel?

-Page 9, line 9: You say that for some days, SO2 is detected by only one instrument because the limit of detection of the other instrument is not exceeded. Is it really true? Did you check that the fact that you consider a circle of 200 km radius around the ground-based station for calculating the IASI average does not play a role (i.e. IASI can detect SO2 in a part of the circle far away from the Brewer)?

-Page 10, line 2: Can you define what is IMO (indicated in Figures 2 and 3)?

-Page 10: Could you specify what are the errors on altitude and SO2 fluxes calculated from ground-based measurements? How were they calculated?

-Page 10, line 15: you say that IASI values can be underestimated. Is it because of low thermal contrast? Did you check the values?

-Page 10, lines 16-18: Since you linearly interpolated the fluxes, you overestimate the total mass when integrating below the red line (especially if the variations of IASI are real). Calculating the total mass on periods where you do not need to interpolate might improve the comparison (when the mass is compared with the one of IASI calculated for the same period).

-Page 10, conclusions: I find the conclusions a bit too short. I would say a word about some of your limitations (underestimation of the masses), about the comparison of the total mass,...

Technical comments

-Abstract, line 4: remove the comma after "data".

-Abstract, lines 9-11: The last sentence is very long and hard to read. You should

rephrase it.

-Page 2, lines 2-5: I think you should rephrase the end of the paragraph, it does not read well.

-Page 2, line 11: I would replace "the first time series of the Holuhraun SO2 plume" by "the first time series of the Holuhraun SO2 emissions".

-Page 2, lines 16-18: I would rephrase the last sentence, it is a little bit too long.

-Page 2, line 21: add "a" before "sampling" and "almost" before "global".

-Page 2, line 26: "are estimate" → "are estimated".

-Page 3, line 30: Replace "The SO2 mass is highest" by "The largest SO2 mass is found".

-Page 6, line 6: "y" should be bold.

-Page 6, line 9: matrixes → matrices.

-Page 7, line 28: "then" → "than".

-Page 8, line 6: Gauthier et al. (2016) is not included in the references at the end of the manuscript.

-Page 8, lines 21-24: I found this sentence very long and difficult to read. I would rephrase it.

-Page 10, line 1: "collated" → "collected"?

-Page 10, line 6: "groud" → "ground".

-Page 10, line 15: "underestimates" → "underestimated".

-Figure 2, top: The blue line is difficult to distinguish from the black dots. Maybe change the colour?

-Figure 3: In the text, you say that some of the ground based measurements provide the altitude of the plume center-of-mass, and the others the altitude of the top of the plume. Maybe you could make the distinction between the two cases in the Figure (circle and triangle, or something else). It would be easier to see which ground-based observations provide the same information than IASI.

References

Bauduin et al., J. Geophys. Res. Atmos. 119, 4253-4263, 2014, doi:10.1002/2013JD021405; Bauduin et al., Atmos. Meas. Tech. 9, 721-740, 2016, doi :10.5194/amt-9-721-2016; Boynard et al., Geophys. Res. Lett. 41, 645-651, 2014, doi :10.1002/2013GL058333; Fioletov et al., Atmos. Chem. Phys. 16, 11497-11519, 2016, doi :10.5194/acp-16-11497-2016; Thordarson and Hartley et al. Geophys. Res. Abstracts, 17 (EGU2015-10708).

---

## Referee Comment (RC2) · Anonymous Referee #2 · 29 May 2018

This paper developed a new scheme to calculate daily SO2 fluxes and average e-folding time for volcanic SO2 emissions in Iceland. In order to overcome the difficulties in latitude and time, the authors propose to use satellite-based thermal infrared spectrometers instead of UV bands to study the volcanic SO2. The results look sound and interesting. I recommend publishing the paper after addressing the comments below.

General comments:

1. Page 3, line 18. In this study all the SO2 measured from 30N to 90N between September 2014 and February 2015 is referred to as Holuhraun SO2. What is the uncertainty of this assumption?

[Figure]

2. This paper is based on the previous work performed by the same author. I understand the authors would like to keep the text simple and avoid repeating contents mentioned by their previous work. However, sometime the text seems to be too brief to keep all important information. For example, Page 3, line 27-28. "regridding the observations of column amount and plume altitude into a 0.125 latitude/longitude boxes following Carboni et al. (2016)." What is special of the regridding approach in Carboni et al. (2016)? I have the similar concern for Section 2.

Specific comments:

1. Page 2, line 22. The exact location of the IASI data should be added.

2. Page 2, line 30. Putting a rough quantification of the uncertainty of the "minimum" here would be appreciated.

3. Page 6, Line 10. The a priori values used were $0.2 \pm 0.2$ Tg/day for flux and $2 \pm 2$ day for the e-folding time. Is there any sources for the priori values? If not, will the fitting results be sensitive to the choices of the priori values?

4. Figure 2. The color of blue is difficult to see.

---

## Author Comment (AC1) · 12 Nov 2018

Please see the attach pdf document

Please also note the supplement to this comment:
https://www.atmos-chem-phys-discuss.net/acp-2018-285/acp-2018-285-AC1-supplement.pdf

---

## Author Comment (AC2) · 12 Nov 2018

Dear ACP Editor and Anonymous Referees,

Please find below our answers to the 2 Anonymous Referees. In blue, the referee's comments, in black our responses.

The manuscript has been improved following the reviewers' requests, in particular note that the effect of clouds above the volcanic plume could significantly change the calculated SO2 total emissions by 50% (from 4.4 Tg to 6.7 Tg).

**Anonymous Referee #1**

In the manuscript entitled "Satellite-derived sulphur dioxide (SO2) emissions from the 2014-2015 Holuhraun eruption (Iceland)", the authors derive the first timeseries of the SO2 emissions for the entire Holuhraun eruption. In a first stage, using a retrieval scheme previously developed, the authors retrieve the SO2 amount and altitude of the Holuhraun plume from IASI observations. Based on these, the authors then determine SO2 total masses every 12 hours in a large box (30\_N-90\_N) covering the Northern Hemisphere. They finally retrieve the SO2 fluxes for 12-hour periods using an optimal estimation scheme, considering the retrieved total masses as the measurement vector.

To assess their results, the authors compare the retrieved SO2 columns, plume altitude and SO2 emissions with different type of ground-based measurements. While the SO2 emissions determined in this paper are important for different applications, a few main issues and several specific comments should be addressed before publication.

Major comments

1) Page 2, lines 27-30, the authors mention that because of some geophysical conditions,

part of the SO2 plume can be missed by IASI, and thus, the derived SO2 masses should be considered as minima. This is totally true, the presence of clouds and/or low thermal contrast can hamper the detection of the SO2, and this is a complicated problem to deal with when estimating the SO2 masses. However, the authors stay very qualitative on this problem and more particularly do not mention this problem anymore in the rest of the paper (e.g. in the comparisons). It seems that the SO2 total mass derived by the authors for the entire eruption is lower than those previously estimated (Gauthier et al, 2016; Pfeffer et al., 2018; Gíslason et al., 2015; Thordarson and Hartley, 2015), but this is not discussed in terms of the underestimation of the SO2 masses and its effect on the estimated fluxes. Since the latter will be available for future comparisons and for model simulations, the authors should discuss this deeper and try to evaluate (and I realize that it is a complicated problem) how large can be the underestimation of the retrieved SO2 masses and how this underestimation affects their SO2 fluxes.

Thank you for this comment. Indeed it is not trivial to estimate the underestimation.

We have now included in the paper a way to estimate how much the SO2 mass could be underestimated due to meteorological cloud above the SO2 plume. This correction can be applied to other datasets that include the altitude of the plume, and is based on monthly cloud statistics from the ESA CCI project. We also included an estimate of SO2 plume missed due to low thermal contrast using the OMI BIRA SO2 dataset.

**Our approach is summarised as follows:**

- We estimated the percent of SO2 missing due to cloud above the plume, as a function of cloud optical depth and the altitude of the meteorological cloud above the SO2 plume, using simulations with a standard atmosphere as done in Fig 6 of Carboni et al 2012: (https://www.atmos-chemphys.net/12/11417/2012/acp-12-11417-2012.pdf)

- Using the ESA cloud CCI dataset of AVHRR (carried on the same platform as IASI and so having the same overpass time) L3 monthly mean statistic, we computed:

1) Monthly mean histograms of frequency of cloud optical depth (COD) at 550 nm,  $\tau$ , averaged over the globe. Cloud optical depth is not present in the cloud L3 database for locations without daylight (e.g. visible channels) and most of the Icelandic plume in the winter months is without daylight, as a consequence here we are assuming the global histogram of frequency of COD is valid over the plume region.

2) Monthly mean histogram of frequency of cloud altitude, averaged on the plume region (30\_N-90\_N). Cloud altitude is available for all locations and during winter months.

We consider the measured mass  $M_{meas}$  to be the difference between true mass  $M_{corr}$  and the missing one  $M_{miss}$ :

$$\begin{split} M_{corr} &- M_{miss} = M_{meas} \\ M_{corr} \left(1 - \frac{M_{miss}}{M_{corr}}\right) = M_{meas} \\ M_{corr} &= M_{meas} \left(\frac{1}{1 - \frac{M_{miss}}{M_{corr}}}\right) \end{split}$$

We compute the correction factor, *C*, for every month of the eruption as a function of altitude, and applied to the vertical distribution dataset.

$$M_{corr}(h) = M_{meas}(h) \cdot C(h)$$

With:

$$C(h) = \frac{1}{(1 - Z(h))}$$

Where Z(h) is the SO2 mass fraction 'missed' in the measurements due to cloud above the plume. Z(h) is estimated as the product of probability of having cloud above altitude *h*, *F*(*h*), times the attenuation due to cloud, *A*,

$$Z(h) = F(h) \cdot A$$

The probability of having cloud above h has been estimate from CCI data for the region considered for the volcanic plume (latitude > 30 N) as the number of cloud retrievals above altitude h divided by number of observations.

Attenuation due to cloud (*A*) is the sum of the frequency of having a cloud with a cloud optical depth  $f(\tau)$  times the attenuation due to a cloud with the same optical depth  $a(\tau)$ .

$$A = \sum_{\tau=0}^{n} f(\tau) a(\tau)$$

 $f(\tau)$  has been estimated using the monthly mean histogram of frequency of cloud optical depth, estimated over the globe.

 $a(\tau)$  has been estimated by running the SO2 retrieval using, as IASI measurements, simulated spectra with water cloud above the plume, using the default atmosphere, and different optical depths at 550 nm. For optical depths bigger than 10 the attenuation is 1 (cloud is opaque and completely mask the SO2 signal).

The figures below show:

- Correction factor.
- SO2 vertical distribution obtained from IASI retrieval (was already in the paper).
- SO2 vertical distribution corrected (for underestimation due to cloud cover).

Figure 1. Correction factor for the  $SO_2$  masses to estimate to correct for the presence of cloud above the plume.